# Quantifying dissipation using fluctuating currents

Junang Li[1], Jordan M. Horowitz[1,2,3], Todd R. Gingrich[1,4] & Nikta Fakhri[1]

Systems coupled to multiple thermodynamic reservoirs can exhibit nonequilibrium dynamics, breaking detailed balance to generate currents. To power these currents, the entropy of the reservoirs increases. The rate of entropy production, or dissipation, is a measure of the statistical irreversibility of the nonequilibrium process. By measuring this irreversibility in several biological systems, recent experiments have detected that particular systems are not in equilibrium. Here we discuss three strategies to replace binary classification (equilibrium versus nonequilibrium) with a quantification of the entropy production rate. To illustrate, we generate time-series data for the evolution of an analytically tractable bead-spring model. Probability currents can be inferred and utilized to indirectly quantify the entropy production rate, but this approach requires prohibitive amounts of data in high-dimensional systems. This curse of dimensionality can be partially mitigated by using the thermodynamic uncertainty relation to bound the entropy production rate using statistical fluctuations in the probability currents.

[1] Department of Physics, Massachusetts Institute of Technology, 77 Massachusetts Avenue, Cambridge, MA 02139, USA. [2] Department of Biophysics, University of Michigan, Ann Arbor, MI 48109, USA. [3] Center for the Study of Complex Systems, University of Michigan, Ann Arbor, MI 48104, USA. [4] Department of Chemistry, Northwestern University, Evanston, IL 60208, USA. Correspondence and requests for materials should be addressed to T.R.G. (email: todd.gingrich@northwestern.edu) or to N.F. (email: fakhri@mit.edu)

Nonequilibrium dynamics is an essential physical feature of biological and active matter systems[1-3]. By harvesting a fuel—in the form of solar energy, a redox potential, or a metabolic sugar—the molecular dynamics in these systems differs profoundly from the equilibrium case. Some of the fuel's free energy is utilized to perform work or is stored in an alternative form, but the remainder is dissipated into the environment, often in the form of heat[1,4]. The energetic loss can alternatively be cast as an increase in entropy of the environment, and the entropy production is associated with broken time-reversal symmetry in the system's dynamics[5-7]. This connection has been leveraged to experimentally classify particular biophysical processes as thermal or active[8,9] based on the existence of probability currents[10,11]. There is great interest in going beyond this binary classification—thermal versus active—to experimentally quantify how active, or how nonequilibrium, a process is[12-14]. Such a quantification could, for example, provide insight into how efficiently molecular motors are able to work together to drive large-scale motions[15-19].

One way to quantify this nonequilibrium activity is to measure the dissipation rate, or how much free energy is lost per unit time. In a biophysical setting, a direct local calorimetric measurement is challenging, but signatures of the dissipation are encoded in stochastic fluctuations of the system[20], even far-from-equilibrium[21-29]. We set out to develop and explore strategies for inferring the dissipation rate from these experimentally-accessible nonequilibrium fluctuations. In a system of interacting driven colloids, where all degrees of freedom are tracked, Lander et al. have indirectly measured dissipation from fluctuations[27]. However, it should also be possible to bound dissipation on the basis of nonequilibrium fluctuations in a subset of the relevant degrees of freedom. As a tangible example of our motivation, consider the experiment of Battle et al., which tracked cilia shape fluctuations to determine that the cilia dynamics were driven out of equilibrium[9]. With suitable analysis of those shape fluctuations, might one determine, or at least constrain, the free energetic cost to sustain the cilia's motion?

Though our ultimate motivations pertain to these complex systems, here we present an exhaustive analytical and numerical study of a tractable model[30]. Using a model consisting of beads coupled by linear springs, we demonstrate how the statistical properties of trajectories provides information about the dissipation rate. The bead-spring model furthermore allows us to address various practical considerations that will be important for future experimental applications of the inference techniques: how much data is required, what is the role of coarse graining, and what can be done about the curse of dimensionality. We show that fluctuations in nonequilibrium currents can provide a route to bound the dissipation rate, even in high-dimensional dynamical systems operating outside a linear-response regime. Crucially, we anticipate many of these insights will support the data analysis of experimentally accessible biological and active matter systems.

## Results

**Bead-spring model.** As one of the simplest possible nonequilibrium models, we consider two coupled beads, each allowed to fluctuate in one dimension. The beads are connected to each other and to the boundary walls by linear springs with stiffness $k$ (see Fig. 1). We imagine that the beads are embedded in two different viscous fluids, one hot with temperature $T_h$ and the other cold with temperature $T_c$. These fluids exert a friction $\gamma$ on each bead, absorbing energy from the beads' motion. In the absence of coupling between the beads, the average rate at which each thermal bath injects energy exactly balances with the rate it absorbs energy due to frictional drag. By coupling the beads, however, there is a net steady-state rate of heat flow $\dot{Q}_{ss}$ from the

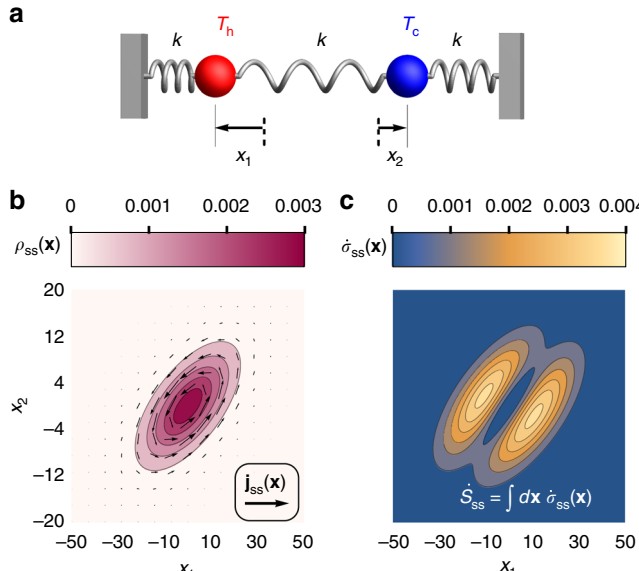

**Fig. 1** Two coupled beads at different temperatures. **a** An illustration of the model with the red bead immersed in a hot temperature bath $T_h$ and the blue bead immersed in a cold temperature bath $T_c$. Three springs with equal spring constant $k$ connected the beads and the walls. Displacements away from the equilibrium position of the hot and cold beads are denoted by $x_1$ and $x_2$, respectively. **b** The steady-state probability density and current as a function of bead displacements for spring constant $k = 1$, friction $\gamma = 1$, and thermal energy scales $k_B T_c = 25$ and $k_B T_h = 250$. **c** The local entropy production rate calculated from Eq. (7) of the system as a function of bead displacements for the same parameters

hot reservoir into the system and out to the cold reservoir. The hot reservoir's entropy changes with rate $\dot{S}_h = -\dot{Q}_{ss}/T_h$ while the cold reservoir's entropy increases with rate $\dot{S}_c = \dot{Q}_{ss}/T_c$. In total, the steady-state entropy production rate can therefore be written as

$$\dot{S}_{ss} = \dot{S}_h + \dot{S}_c = \dot{Q}_{ss}(T_c^{-1} - T_h^{-1}). \tag{1}$$

This equation expresses the entropy production rate as the product of a flux $\dot{Q}_{ss}$ and the conjugate thermodynamic driving force $(T_c^{-1} - T_h^{-1})$. The typical situation is that the driving force may be tuned in the lab and the flux is measured as a response.

Suppose, however, that it is not simple to measure the heat flux. Rather, we imagine directly observing the bead positions as a function of time. Those measurements are sufficient to extract the entropy production rate, but to do so we must go beyond the thermodynamics and explicitly consider the system's dynamics, an approach known as stochastic thermodynamics[1,31,32]. The starting point is to mathematically describe the bead-spring dynamics with a coupled overdamped Langevin equation $\dot{\mathbf{x}} = A\mathbf{x} + F\boldsymbol{\xi}$, where $\mathbf{x} = (x_1, x_2)^T$ is the vector consisting of each bead's displacement from its equilibrium position, $\boldsymbol{\xi} = (\xi_1, \xi_2)^T$ is a vector of independent Gaussian white noises, and

$$A = \begin{pmatrix} -2k/\gamma & k/\gamma \\ k/\gamma & -2k/\gamma \end{pmatrix}, \quad F = \begin{pmatrix} \sqrt{2k_B T_h/\gamma} & 0 \\ 0 & \sqrt{2k_B T_c/\gamma} \end{pmatrix}. \tag{2}$$

The matrix $A$ captures deterministic forces acting on the beads due to the springs, while $F$ describes the random forces imparted by the medium. The strength of these random forces depends on

the temperature and the Boltzmann constant $k_B$, consistent with the fluctuation-dissipation theorem[33].

It is useful to cast the Langevin equation as a corresponding Fokker-Planck equation for the probability of observing the system in configuration $\mathbf{x}$ at time $t$, $\rho(\mathbf{x}, t)$:

$$\frac{\partial \rho(\mathbf{x}, t)}{\partial t} = -\nabla \cdot (A\mathbf{x}\rho(\mathbf{x}, t) - D\nabla\rho(\mathbf{x}, t)) \equiv -\nabla \cdot \mathbf{j}(\mathbf{x}, t), \quad (3)$$

with $D = FF^T/2$. Though we are modeling a two-particle system, it can be helpful to think of the entire system as being a single point diffusing through $\mathbf{x}$ space with diffusion tensor $D$ and with deterministic force $\gamma A\mathbf{x}$. The second equality in Eq. (3) defines the probability current $\mathbf{j}(\mathbf{x}, t)$. These probability currents (and their fluctuations) will play a central role in our strategies for inferring the rate of entropy production.

Due to its analytic and experimental tractability, this bead-spring system and related variants have been extensively studied as models for nonequilibrium dynamics[34–39]. In particular, the steady-state properties are well-known. Correlations between the position of bead $i$ at time 0 and that of bead $j$ at time $t$ are given by $C_{ij}(t) = \langle x_i(0)x_j(t)\rangle$. The expectation value is taken over realizations of the Gaussian noise to give

$$C(t) = \int_{-\infty}^{t} ds\, e^{A(t-s)} FF^T e^{A^T(t-s)}. \quad (4)$$

The steady-state density and current are expressed simply as

$$\begin{aligned} \rho_{ss}(\mathbf{x}) &= (2\pi\sqrt{\det\mathcal{C}})^{-1} e^{-\frac{1}{2}\mathbf{x}^T \mathcal{C}^{-1}\mathbf{x}} \\ \mathbf{j}_{ss}(\mathbf{x}) &= (A\mathbf{x} + D\mathcal{C}^{-1}\mathbf{x})\rho_{ss}(\mathbf{x}) \end{aligned} \quad (5)$$

in terms of the long-time limit of the correlation matrix

$$\mathcal{C} \equiv \lim_{t\to\infty} C(t) = \frac{k_B}{12k}\begin{pmatrix} 7T_h + T_c & 2(T_c + T_h) \\ 2(T_c + T_h) & T_h + 7T_c \end{pmatrix}. \quad (6)$$

The steady-state current $\mathbf{j}_{ss}(\mathbf{x})$ is a vector field that specifies the probability current conditioned upon the system being in configuration $\mathbf{x}$. Associated with this current is a local conjugate thermodynamic force $\mathbf{F}(\mathbf{x}) = k_B \mathbf{j}_{ss}^T(\mathbf{x}) D^{-1}/\rho_{ss}(\mathbf{x})$[40,41]. The product of the microscopic current and force is the local entropy production rate at configuration $\mathbf{x}$: $\dot{\sigma}_{ss}(\mathbf{x}) = \mathbf{F}(\mathbf{x}) \cdot \mathbf{j}_{ss}(\mathbf{x})$. Upon integrating over all configurations, we obtain the total entropy production rate

$$\begin{aligned} \dot{S}_{ss} &= \int d\mathbf{x}\, \dot{\sigma}_{ss}(\mathbf{x}) = k_B \text{Tr}\{AD^{-1}A\mathcal{C} - \mathcal{C}^{-1}D\} \\ &= k_B \frac{k(T_h - T_c)^2}{4\gamma T_h T_c}. \end{aligned} \quad (7)$$

Comparing with Eq. (1), we see that the rate of net heat flow is $\dot{Q}_{ss} = k_B k(T_h - T_c)/4\gamma$. Our ability to analytically compute the heat flow derives from the linear coupling between beads, yet we are ultimately interested in experimental scenarios in which linear coupling could not be assumed. In those more complicated systems, there is no simple analytical expression for the local entropy production rate, but we could still estimate $\dot{\sigma}_{ss}$ by sampling trajectories from the steady-state distributions—either in a computer or in the lab. We now consider strategies for this estimation by sampling the bead-spring dynamics and comparing with the analytical expression, Eq. (7).

**Estimating the steady state from sampled trajectories**. We first seek estimates of $\mathbf{j}_{ss}(\mathbf{x})$ and $\rho_{ss}(\mathbf{x})$ from a long trajectory $\mathbf{x}(t)$ of bead positions over an observation time $\tau_{obs}$. We estimate the steady-state density by the empirical density, the fraction of time the trajectory spends in state $\mathbf{x}$:

$$\rho(\mathbf{x}) = \frac{1}{\tau_{obs}} \int_0^{\tau_{obs}} \delta(\mathbf{x}(t) - \mathbf{x}) dt, \quad (8)$$

where $\delta$ is a Dirac delta function. The empirical density is an unbiased estimate of the steady-state density, meaning the fluctuating density $\rho(\mathbf{x})$ tends to $\rho_{ss}(\mathbf{x})$ in the long-time limit. Similarly, an unbiased estimate for the steady-state currents is the empirical current

$$\mathbf{j}(\mathbf{x}) = \frac{1}{\tau_{obs}} \int_0^{\tau_{obs}} \delta(\mathbf{x}(t) - \mathbf{x}) \circ d\mathbf{x}(t). \quad (9)$$

This Stratonovich integral can be colloquially read as the time-average of all displacement vectors that were observed when the system occupied configuration $\mathbf{x}$. In practice, experiments typically record the configuration $\mathbf{x}$ at discrete-time intervals $\Delta t$ such that the trajectory is given by the timeseries $\{\mathbf{x}_{\Delta t}, \mathbf{x}_{2\Delta t}...\}$. Consequently we work with estimates of the density and currents[42]:

$$\hat{\rho}(\mathbf{x}) = \frac{\Delta t}{\tau_{obs}} \sum_{i=1}^{\tau_{obs}/\Delta t} K(\mathbf{x}_{i\Delta t}, \mathbf{x}) \quad (10)$$

$$\hat{\jmath}(\mathbf{x}) = \frac{\hat{\rho}(\mathbf{x})}{2\Delta t} \frac{\sum_{i=2}^{\tau_{obs}/\Delta t - 1} L(\mathbf{x}_{i\Delta t}, \mathbf{x})\left[\mathbf{x}_{(i+1)\Delta t} - \mathbf{x}_{(i-1)\Delta t}\right]}{\sum_{i=2}^{\tau_{obs}/\Delta t - 1} L(\mathbf{x}_{i\Delta t}, \mathbf{x})}, \quad (11)$$

where $K$ and $L$ are kernel functions[43]. The kernel functions make it natural to spatially coarse grain the data, a necessity because experiments have a limited resolution and because most microscopic configurations will never be sampled by a finite-length trajectory. The function $K(\mathbf{x}_{i\Delta t}, \mathbf{x})$ controls how observing the $i$th data point at position $\mathbf{x}_{i\Delta t}$ impacts the estimate of $\hat{\rho}$ at a nearby position $\mathbf{x}$. Similarly, $L$ controls how currents are estimated in the neighborhood of the observed data points. Specific choices for $K$ and $L$ are discussed in the Methods section. Using $\hat{\rho}$ and $\hat{\jmath}$ we can now construct direct estimates of the entropy production rate.

**Direct strategies for entropy production inference**. In computing Eq. (7), we integrated the local entropy production rate $\mathbf{F}(\mathbf{x}) \cdot \mathbf{j}_{ss}(\mathbf{x})$ over all configurations $\mathbf{x}$. When $\mathbf{j}_{ss}(\mathbf{x})$ and $\mathbf{F}(\mathbf{x})$ are not known, it is natural to replace them by the estimators $\hat{\jmath}(\mathbf{x})$ and $\hat{\mathbf{F}}(\mathbf{x}) \equiv k_B \hat{\jmath}^T(\mathbf{x}) D^{-1}/\hat{\rho}(\mathbf{x})$. Though $\hat{\mathbf{F}}$ is constructed from unbiased estimators $\hat{\jmath}$ and $\hat{\rho}$, $\hat{\mathbf{F}}$ is only asymptotically unbiased, necessitating sufficiently long trajectories for the bias to become negligible. Utilizing $\hat{\mathbf{F}}$, we approximate $\dot{S}_{ss}$ by either a spatial or a temporal average:

$$\widehat{S}_{ss}^{spat} \equiv \int d\mathbf{x}\, \hat{\mathbf{F}}(\mathbf{x}) \cdot \hat{\jmath}(\mathbf{x}) \quad (12)$$

$$\begin{aligned} \widehat{S}_{ss}^{temp} &\equiv \frac{1}{\tau_{obs}} \int_0^{\tau_{obs}} dt\, \hat{\mathbf{F}}(\mathbf{x}(t)) \cdot \circ d\mathbf{x}(t) \\ &= \frac{1}{\tau_{obs}} \sum_{i=2}^{\tau_{obs}/\Delta t} \hat{\mathbf{F}}\left(\frac{\mathbf{x}_{i\Delta t} + \mathbf{x}_{(i-1)\Delta t}}{2}\right) \cdot \left[\mathbf{x}_{i\Delta t} - \mathbf{x}_{(i-1)\Delta t}\right]. \end{aligned} \quad (13)$$

The performance of these estimators is assessed using data sampled from numerical simulations of the Langevin equation, described further in Methods. As illustrated in Fig. 2, the estimators are biased for any finite trajectory length, but they converge to the analytical result, Eq. (7), with sufficiently long sampling times.

At first glance $\widehat{S}_{ss}^{spat}$ and $\widehat{S}_{ss}^{temp}$ may appear equivalent due to ergodicity. Indeed, with an infinite amount of sampling, both schemes must yield the same result, $\dot{S}_{ss}$, but the temporal estimator converges significantly faster with finite sampling. Plots of the estimated local dissipation rate (Fig. 2 inset) hint at the reason $\widehat{S}_{ss}^{spat}$ converges more slowly: $\dot{\sigma}_{ss}(\mathbf{x})$ must be accurately estimated by $\hat{\sigma}_{ss}(\mathbf{x}) = \hat{\mathbf{F}}(\mathbf{x}) \cdot \hat{\jmath}(\mathbf{x})$ throughout the entire

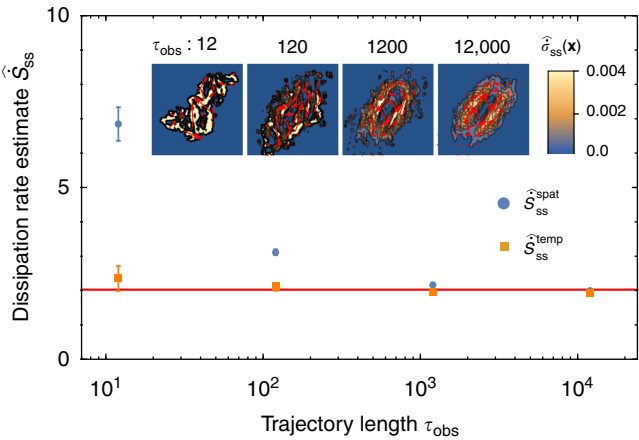

**Fig. 2** Convergence of dissipation estimates. The spatial (blue solid circles) and temporal (red solid squares) dissipation rate estimates converge to the steady-state value $\dot{S}_{ss}$ (red line) of Eq. (7). Estimates of the total dissipation rate, calculated from Eqs. (12) and (13), are extracted from Langevin trajectories simulated for time $\tau_{obs}$ with timestep $10^{-3}$ using $k = \gamma = 1$, $k_B T_c = 25$, and $k_B T_h = 250$ as in Fig. 1. Error bars are the standard error of the mean, computed from 10 independent trajectories. Estimates of the local dissipation rate from the spatial estimator with different trajectory lengths are plotted in the inset

configuration space. The integral in Eq. (12) equally weights $\hat{\sigma}_{ss}(\mathbf{x})$ at all $\mathbf{x}$, even those points which have been infrequently (or never) visited by the stochastic trajectory. Our $\mathbf{x}$ has dimension two, but we will also consider higher-dimensional configuration spaces, for example by coupling more than two beads in a linear chain. If that configuration space has dimension greater than three or four, it becomes impractical to estimate $\dot{\sigma}_{ss}$ across the entire space. Furthermore, estimating Eq. (12) for high-dimensional $\mathbf{x}$ confronts the classic problem of performing numerical quadrature on a high-dimensional grid, where it is well-known that Monte Carlo integration becomes a superior method.

The temporal integral can be thought of as a convenient way to implement such a Monte Carlo integration, with sampled $\mathbf{x}$'s coming from the configurations of the stochastic trajectory. Notably, Eq. (13) is computed from estimates of the thermodynamic force near the sampled configurations $\mathbf{x}_{i\Delta t}$, precisely where the finite trajectory has been most reliably sampled. In contrast, Eq. (12) requires spurious extrapolation of the kernel density estimates ($\hat{\rho}$ and $\hat{j}$) to points which are far from the any sampled configurations. The advantage of the temporal estimator over the spatial one becomes even more pronounced as dimensionality increases. Nevertheless, even $\hat{S}_{ss}^{temp}$ becomes harder to estimate when $\mathbf{x}$ grows in dimensionality. Getting accurate estimates of $\mathbf{F}$ around the $\mathbf{x}_{i\Delta t}$ requires observing several trajectories which have cut through that part of configuration space while traveling in each direction. But when the dimensionality is large, recurrence to the same configuration-space neighborhood takes a long time. Consequently, we turn to a complementary method which can be informative even when $\mathbf{x}$ is too high-dimensional to accurately estimate $\mathbf{F}$.

**Indirect strategy for entropy production inference**. Thus far our estimators have been based on detailed microscopic information, but as the dimensionality of $\mathbf{x}$ increases, estimating the microscopic steady-state properties requires exponentially more data. To combat this curse of dimensionality, it is standard to project

high-dimensional dynamics onto a few preferred degrees of freedom[9,44–46]. For example, the projected coordinates could be two principle components from a principle component analysis. Such projected dynamics have been used to detect broken detailed balance[9], however, these reduced dynamics overlook hidden dissipation from the discarded degrees of freedom.

An alternative strategy that retains contributions from all degrees of freedom is provided by recent theoretical results relating entropy production and current fluctuations in general nonequilibrium steady-state dynamics[28,29,47–52]. To this end, we introduce a single projected macroscopic current, constructed as a linear combination of the microscopic currents:

$$j_{\mathbf{d}} = \int d\mathbf{x}\, \mathbf{d}(\mathbf{x}) \cdot \mathbf{j}(\mathbf{x}), \qquad (14)$$

where $\mathbf{d}(\mathbf{x})$ is a vector field that weights how much a microscopic current at $\mathbf{x}$ contributes to the macroscopic current $j_{\mathbf{d}}$. Any physically measurable current—electrons flowing through a wire, heat passing from one bead to the other, or the production of a chemical species in a reaction network—can be cast as such a linear superposition of microscopic currents. Figure 3 illustrates one particular example by applying the weighting field $\mathbf{d}(\mathbf{x}) = \mathbf{F}(\mathbf{x})$ to project microscopic currents onto the single macroscopic current $j_{\mathbf{F}}$. Each step of the trajectory is weighted by the value of $\mathbf{d}$ associated with the observed transition, and this weighted average, accumulated as a function of time, is the fluctuating macroscopic current (fluctuating because it depends on the particular stochastic trajectory). Each trajectory observed for a time $\tau_{obs}$ yields a measurement $j_{\mathbf{d}}$ of the fluctuating current, and many such trajectories give a distribution $P(j_{\mathbf{d}})$ characterized by mean $\langle j_{\mathbf{d}} \rangle$ and variance $\text{Var}(j_{\mathbf{d}})$. The thermodynamic uncertainty relation (TUR)[28,29,48–50] then constrains the entropy production rate in terms of the dynamical fluctuations of this macroscopic current:

$$\dot{S}_{ss} \geq \frac{2 k_B \langle j_{\mathbf{d}} \rangle^2}{\tau_{obs} \text{Var}(j_{\mathbf{d}})} \equiv \dot{S}_{TUR}^{(\mathbf{d})}. \qquad (15)$$

Note that we have used $\text{Var}(j_{\mathbf{d}})$ to denote the variance of the macroscopic empirical current distribution, but some prior work[29,48] used this notation to denote the way the variance scaled with observation time. The difference between these notations is the factor of $\tau_{obs}$ in the denominator of the right hand side of Eq. (15).

Unlike the field of microscopic currents, $\mathbf{j}(\mathbf{x})$, the macroscopic current $j_{\mathbf{d}}$ is a single scalar quantity, allowing estimates of its cumulants—particularly the mean $\widehat{\langle j_{\mathbf{d}} \rangle}$ and variance $\widehat{\text{Var}(j_{\mathbf{d}})}$—to be extracted from a modest amount of experimental data. Indeed, measurements of kinesin fluctuations have recently been used to infer constraints on the efficiency of these molecular motors[18,53]. Importantly, the TUR is valid for any choice of $\mathbf{d}$, granting freedom to consider fluctuations of arbitrary macroscopic currents, some of which will yield tighter bounds than others. In a later section, we use Monte Carlo sampling to seek a choice for $\mathbf{d}$ which yields the tightest possible bound, but first we consider an important physically motivated choice, $\mathbf{d} = \mathbf{F}$. In this case, the macroscopic current $j_{\mathbf{F}}$ is the fluctuating entropy production rate (cf. Eqs. (7) and (14)), so $\langle j_{\mathbf{F}} \rangle = \dot{S}_{ss}$. With access to $\mathbf{F}$, we can thus compute the entropy production rate by simply taking the mean of the generalized current (the average slope in Fig. 3), or we could use the fluctuations from repeated realizations of $j_{\mathbf{F}}$ to get a bound on $\dot{S}_{ss}$ via Eq. (15).

It perhaps seems foolish to settle for a bound if one could compute the actual entropy production rate, but in practice one would not typically have access to $\mathbf{F}$. More likely, it would only be

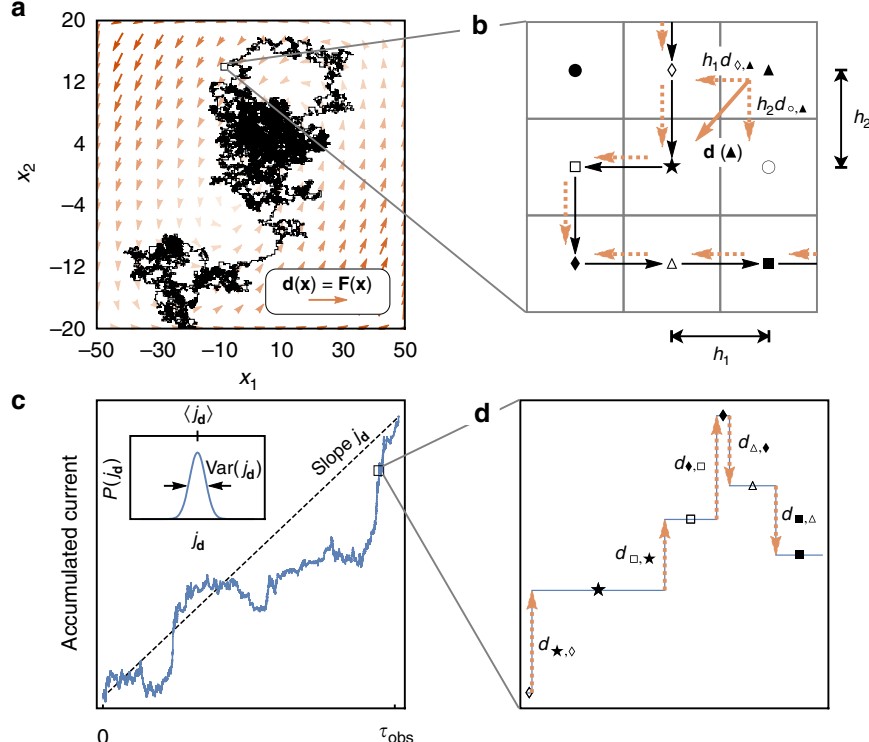

**Fig. 3** Extracting current fluctuations from trajectories. **a** A realization of a long trajectory diffusing through configuration space. The macroscopic current is computing by choosing a vector field **d**(**x**), here chosen as the thermodynamic force field **F**(**x**). **b** On the microscopic scale, the trajectory may be modeled as discrete jumps between neighboring lattice sites (here labeled with symbols: lozenge, star, square, ...). The continuous-space vector field is decomposed into components along the direction of possible jumps, i.e., **d** evaluated at the black triangle site can be expressed in terms of the weight $d_{\diamond, \blacktriangle}$ associated with a jump from the black triangle to the white diamond. **c** A realization of the trajectory gives a single value of the empirical current $j_{\mathbf{d}}$. By recording many realizations, the empirical current distribution $P(j_{\mathbf{d}})$ is sampled to give $\langle j_{\mathbf{d}} \rangle$ and $\text{Var}(j_{\mathbf{d}})$ (inset). In the case that $\mathbf{d} = \mathbf{F}$, the mean slope of this accumulated current is the average entropy production rate. **d** The empirical current for a single realization is constructed as the sum of the **d** weights for each microscopic transition of the jump process

possible to estimate **F** from data as $\hat{\mathbf{F}}$. With sufficient data, $\hat{\mathbf{F}}$ converges to **F** such that a temporal estimate of the entropy production rate would eventually become accurate, but this convergence is slow in high dimensions. Alternatively, by choosing $\mathbf{d} = \hat{\mathbf{F}}$, we obtain a TUR lower bound estimate

$$\widehat{\dot{S}}_{\text{TUR}}^{(\hat{\mathbf{F}})} = \frac{2k_{\text{B}} \widehat{\langle j_{\hat{\mathbf{F}}} \rangle}^2}{\tau_{\text{obs}} \widehat{\text{Var}(j_{\hat{\mathbf{F}}})}}. \tag{16}$$

A key advantage of this estimate is that it is less sensitive to whether $\hat{\mathbf{F}}$ has converged than either $\widehat{\dot{S}}_{\text{ss}}^{\text{spat}}$ and $\widehat{\dot{S}}_{\text{ss}}^{\text{temp}}$. When $\hat{\mathbf{F}}$ is noisily estimated due to little data or high dimensionality, the TUR estimate can nevertheless provide an accessible route to constraining the entropy production rate from experimental data.

**Convergence of the entropy production rate estimates**. To assess the costs and benefits of the various estimation schemes, we numerically sampled trajectories for the two-bead model of Fig. 1 and for a variant with five beads coupled along a one-dimensional chain with spring constant $k$, the five beads being embedded in thermal baths whose temperatures linearly ramp from $T_{\text{c}}$ to $T_{\text{h}}$. Equation (7) gives the entropy production rate for the two-bead model as a function of the bath temperatures. An analogous expression is derived in Supplementary Note 1 for the model with five beads, and both expressions are plotted with a solid red line in Fig. 4. The temporal and spatial estimators both converge to these analytical expressions in the long

trajectory limit, while the TUR estimate tends to the lower bound $\dot{S}_{\text{TUR}}^{(\mathbf{d})}$. We performed a series of calculations to assess: (1) how close is this lower bound to the true dissipation rate and (2) how long of a trajectory is needed to converge all three estimates.

We discuss the convergence results first, plotted as insets in Fig. 4. Using a trajectory of length $\tau_{\text{obs}}$, $\hat{\mathbf{F}}$ was estimated, and this estimated thermodynamic force field was used to plot how quickly $\widehat{\dot{S}}_{\text{ss}}^{\text{spat}}$ and $\widehat{\dot{S}}_{\text{ss}}^{\text{temp}}$ converged to their expected value of $\dot{S}_{\text{ss}}$. To compare convergence of the TUR bound on an equal footing, we recognize that the $\tau_{\text{obs}} \to \infty$ limit of a long trajectory with perfect sampling will not yield $\dot{S}_{\text{ss}}$ but rather the bound $\dot{S}_{\text{TUR}}^{(\mathbf{F})}$. In all three cases we scale the estimate by its appropriate infinite-sampling limit and observe how quickly this ratio decays to one. The superiority of the temporal estimator over the spatial one is clear in the two-bead model, and the inadequacy of the spatial estimator is so stark in the higher-dimensional five-bead model that it was prohibitive to compute. The TUR estimator performance is comparable to the temporal average estimator when **F** can be estimated well (low dimensionality and large thermodynamic driving). In the more challenging situation that the phase space is high dimensional and the statistical irreversibility is more subtle, the TUR estimator appears to offer some advantage. It converges with roughly an order of magnitude fewer samples than are required for $\widehat{\dot{S}}_{\text{ss}}^{\text{temp}}$ (see bottom right inset of Fig. 4b).

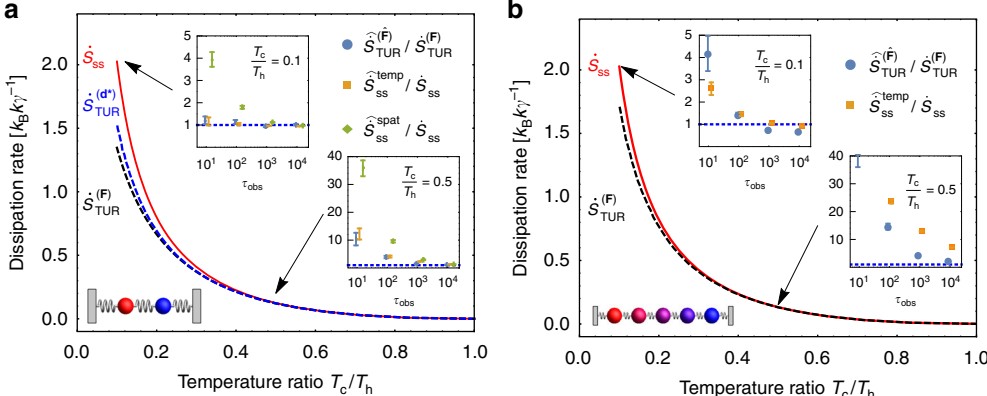

**Fig. 4** Performance of dissipation rate estimators. Data are shown both for the model with two beads (**a**) and the higher-dimensional model with five beads (**b**). The TUR bound with $\mathbf{d} = \mathbf{F}$ (dashed black line) becomes tighter to the actual dissipation rate (solid red line) when the dynamics is closer to equilibrium ($T_c/T_h \to 1$) and in the limit of many beads. Inset plots show the estimator convergence rates for temperature ratios of 0.1 and 0.5, with error bars reporting standard error, computed from 10 independent samples. The blue dashed line in **a** is the TUR bound resulting from a Monte Carlo optimization scheme, as described further in Fig. 5. The bottom right inset of **b** reflects that the TUR estimator may be useful as a practical proxy for the entropy production rate for high-dimensional systems when the dynamics is weakly driven

To understand how well one can estimate the entropy production rate from current fluctuations, we must also address how close the TUR lower bound is to $\dot{S}_{ss}$. The dashed black line of Fig. 4 shows that the TUR lower bound equals the actual entropy production rate in the near-equilibrium limit $T_c \to T_h$. Far from equilibrium, the TUR lower bound remains the same order of magnitude as the entropy production rate, with the deviation increasing with the size of the temperature difference. Comparing the dashed black lines in two different dimensions, we can see that as more beads are added to the model, this deviation between $\dot{S}_{TUR}^{(\mathbf{F})}$ and $\dot{S}_{ss}$ decreases. Hence the TUR bound more closely approximates the actual entropy production rate with increasing dimensionality and decreasing thermodynamic force, precisely the conditions when the TUR estimate converges more rapidly.

**Optimizing the macroscopic current**. Thus far we have focused on measuring the statistics of a particular macroscopic empirical current, the fluctuating entropy production, constructed by choosing $\mathbf{d} = \mathbf{F}$. This choice was a natural starting point since the fluctuations are known to saturate Eq. (15) in the equilibrium limit $T_c \to T_h$[29]. However, when working with timeseries data we had to replace $\mathbf{F}$ by the estimate $\hat{\mathbf{F}}$, and this estimated thermodynamic force is error-prone in high dimensions. In the previous section we saw that the TUR estimator is sufficiently robust that a tight bound for $\dot{S}_{ss}$ may be inferred even when $\hat{\mathbf{F}}$ has not fully converged to $\mathbf{F}$. This robustness derives from the validity of Eq. (15) for all possible choices of $\mathbf{d}$. The generality of the TUR can be further leveraged by optimizing over $\mathbf{d}$:

$$\dot{S}_{ss} \geq \frac{2k_B}{\tau_{obs}} \max_{\mathbf{d(x)}} \frac{\langle j_{\mathbf{d}} \rangle^2}{\mathrm{Var}(j_{\mathbf{d}})}. \tag{17}$$

We are not aware of methods to explicitly compute the optimal choice of $\mathbf{d}$, but a vector field $\mathbf{d}^*(\mathbf{x})$ which outperforms $\mathbf{F}(\mathbf{x})$ can be found readily by Monte Carlo (MC) sampling with a preference for macroscopic currents with a large TUR ratio $\langle j_{\mathbf{d}} \rangle^2 / \mathrm{Var}(j_{\mathbf{d}})$.

Each step of the MC algorithm requires $\langle j_{\mathbf{d}} \rangle$ and $\mathrm{Var}(j_{\mathbf{d}})$, which could be estimated with trajectory sampling, as illustrated in Fig. 3a, c. In fact, one could collect a single long trajectory—from an experiment or from simulation—then sample $\mathbf{d}^*$ based on mean

and variance estimates $\widehat{\langle j_{\mathbf{d}^*} \rangle}$ and $\langle \widehat{\mathrm{Var}(j_{\mathbf{d}^*})} \rangle$ for that fixed trajectory. Such a scheme is enticing, but we warn that the procedure is susceptible to over-optimization of the TUR ratio since optimizing to maximize the ratio $\widehat{\langle j_{\mathbf{d}^*} \rangle}^2 / \widehat{\mathrm{Var}(j_{\mathbf{d}^*})}$ is not the same as optimizing the ratio $\langle j_{\mathbf{d}^*} \rangle^2 / \mathrm{Var}(j_{\mathbf{d}^*})$. The former can yield a large value just because the trajectory happens to return anomalous estimates for the mean and variance of the generalized current. The latter ratio does not depend on any one trajectory but has rather averaged over all trajectories. Avoiding over-optimization requires appropriate cross-validation. For example, $\mathbf{d}^*$ could be selected based on one sampled trajectory then the dissipation bound inferred by an independently sampled trajectory.

Rather than implementing such a cross-validation scheme, we avoided the over-optimization problem for this model system by putting the dynamics on a grid to compute the means and variances exactly. As described in Methods, we construct a continuous-time Markov jump process on a square lattice with grid spacing $\mathbf{h} = \{h_1, h_2\}$ such that the $\mathbf{h} \to 0$ jump process limits to the same Fokker-Planck description, Eq.(3), as the continuous-space Langevin dynamics[48]. The vector field $\mathbf{d}(\mathbf{x})$ is also discretized as a set of weights $d_{\mathbf{x+h,x}}$ associated with the transition from $\mathbf{x}$ to the neighboring microstate at $\mathbf{x} + \mathbf{h}$ (see Fig. 3b, d). In place of trajectory sampling, the mean and variance can be extracted from a standard computation of the current's scaled cumulant generating function as a maximum eigenvalue of a tilted rate matrix[54–56].

The MC sampling returns an ensemble of nearly-optimal choices for $\mathbf{d}^*$ such that $\dot{S}_{ss} \geq \dot{S}_{TUR}^{(\mathbf{d}^*)} \geq \dot{S}_{TUR}^{(\mathbf{F})}$. Each $\mathbf{d}^*$ from the ensemble yields a similar TUR ratio, but the near-optimal vector fields are qualitatively distinct (see Fig. 5). We lack a physical understanding of the differences between the various near-optimal choices $\mathbf{d}^*$ and the thermodynamic force field $\mathbf{F}$. Even without a clear physical interpretation, we have a straightforward numerical procedure for extracting as tight of an entropy production bound as can be obtained from macroscopic current fluctuations.

## Discussion
Physical systems in contact with multiple thermodynamic reservoirs support nonequilibrium dynamics that manifest as probability currents in phase space. Detection of these currents has

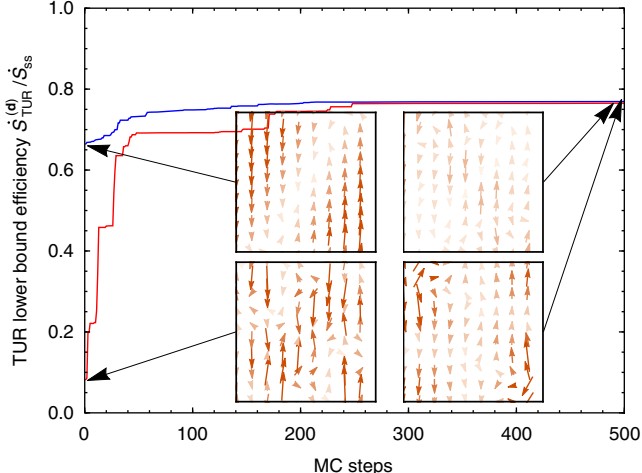

**Fig. 5** Monte Carlo sampling for maximally informative currents. We seek a weighting vector field **d** such that the TUR bound is as close to the true entropy production rate as possible. Starting either with **d** = **F** (blue curve) or with a random vector field (red curve), a Markov chain Monte Carlo procedure was used to change **d** in search of a higher $\dot{S}_{TUR}^{(d)}/\dot{S}_{ss}$ ratio. The Monte Carlo sampling discovers diverse ways to yield a similar maximal value of the TUR ratio, suggesting that while the optimization problem is not dominated by a single basin, competitive near-optimal solutions can be discovered from a variety of starting points

been used in a biophysical context to differentiate between dissipative and equilibrium processes. In this paper, we have explored how the currents can be further utilized to infer the rate of entropy production. Using a solvable toy model, we demonstrated three inference strategies: one based on a spatial average, one based on a temporal average, and one based on fluctuations in the currents.

Regardless of strategy, the entropy production inference becomes more challenging and requires more data as the thermodynamic drive decreases. This challenge results from the fact that weakly driven systems produce trajectories which look very similar when played forward or backward in time. The weaker the drive, the more data it requires to confidently detect the statistical irreversibility.

It is in this weak driving limit that we see the most stark difference between the performance of the three studied estimators. As we move to higher-dimensional but weakly driven systems, it requires too much data to detect the statistical irreversibility at every point in phase space, so performing spatial averages is out of the question. The temporal average can still be taken, but for a fixed amount of data, estimates of **F** become systematically more error-prone with increased dimensionality. In that limit we find it useful to measure not just the average current, but also the variance. By leveraging the TUR we circumvent the need to accurately estimate **F** and achieve more rapid convergence.

The TUR-inspired estimator is not without pitfalls. Most prominently, it only returns a bound on the entropy production rate, and it is not simple to understand how tight this bound will be. That tightness, characterized by $\eta \equiv \dot{S}_{TUR}^{(F)}/\dot{S}_{ss}$, does not, for example, depend solely on the strength of the thermodynamic drive. In Supplementary Note 2 and Supplementary Figure 2, we make this point by separately tuning the various spring constants to show how $\eta$ depends on properties of the system in addition to the ratio of reservoir temperatures. Though our modestly sized toy systems (no more than five coupled beads) always produce $\eta$ of order unity, there is little

reason to believe that the TUR bound will remain a good proxy for the entropy production rate in the limit of a high-dimensional system in which only a few degrees of freedom are visible. Future experiments are needed to elucidate whether these inference strategies can be usefully applied to the complex biophysical dynamics that has motivated our study.

## Methods

**Numerically generating the bead-spring dynamics**. We simulate the bead-spring dynamics in two complementary ways: as discrete-time trajectories in continuous-space and as continuous-time trajectories in discrete space. The results presented in Figs. 2 and 4 stem from continuous-space calculations. Trajectories are generated by numerically integrating the overdamped Langevin equation using the stochastic Euler integrator with timestep $\Delta t$ according to $\mathbf{x}_{(i+1)\Delta t} = \mathbf{x}_{i\Delta t} + A\mathbf{x}_{i\Delta t}\Delta t + F\boldsymbol{\eta}$, where $\boldsymbol{\eta}$ is a vector of random numbers drawn from the normal distribution with variance $\Delta t$ for each timestep. Setting $k = \gamma = 1$, we numerically integrate the equation of motion with timestep $\Delta t = 0.001$. The initial condition $\mathbf{x}_0$ is effectively drawn from the steady state by starting the clock after integrating the dynamics for a long time from a random initial configuration. In addition to the discrete-time simulations, continuous-time jump trajectories were simulated in discrete space with a rate

$$\mathbb{W}_{\mathbf{x}+\mathbf{h},\mathbf{x}} = \left[ (A\mathbf{x}/2) + \mathbf{h}^T D \right] \cdot \mathbf{h}/\mathbf{h}^T \mathbf{h} \qquad (18)$$

for transitioning from a lattice site at position $\mathbf{x}$ to a neighboring site at position $\mathbf{x} + \mathbf{h}$[48]. This discrete-space trajectory was generated by first discretizing the phase space on a 200 by 200 grid with $x_1$ ranging from $-50$ to $50$ and $x_2$ ranging from $-20$ to $20$ as shown in Fig. 3a. The Markov jump process is simulated using the Gillespie algorithm[57].

**Estimating density and current**. To form histogrammed estimates, we bin the data into a 100 by 100 grid with $x_1$ ranging between $\pm50$ and $x_2$ ranging between $\pm20$. We can write the kernel functions as $K(\mathbf{x}_{i\Delta t}, \mathbf{x}) = L(\mathbf{x}_{i\Delta t}, \mathbf{x}) = \sum_{m,n} \chi_{mn}(\mathbf{x})\chi_{mn}(\mathbf{x}_{i\Delta t})$, where $\chi_{mn}$ is the indicator function taking the value 1 only if the argument lies in the bin with row and column indices $m$ and $n$. Alternatively, a continuous estimate of the density and current can be constructed using smooth non-negative functions for $K$ and $L$, each of which integrates to one. For our kernel density estimates, we place a Gaussian at each data point by choosing $K(\mathbf{x}_{i\Delta t}, \mathbf{x}) \propto \exp[(\mathbf{x} - \mathbf{x}_{i\Delta t})^T \Sigma^{-1}(\mathbf{x} - \mathbf{x}_{i\Delta t})]$. The breadth of the $i$th Gaussians $b_i$, known as the bandwidth, sets the diagonal matrix $\Sigma^{-1}$ via $\Sigma_{ii} = b_i^2$. The estimation of currents proceeds similarly using kernel regression with the Epanechnikov kernel[58]

$$L(\mathbf{x}_{i\Delta t}, \mathbf{x}) \propto \begin{cases} \prod_{j=1}^{d}\left(1 - \frac{(x_{i\Delta t;j} - x_j)^2}{b_j^2}\right), & |\mathbf{x}_{i\Delta t} - \mathbf{x}| < \mathbf{b} \\ 0, & \text{otherwise,} \end{cases} \qquad (19)$$

where $d$ is the spatial dimension and $x_{i\Delta t;j}$ is the $j$th component of the configuration $\mathbf{x}_{i\Delta t}$ at discrete time $i$. The bandwidth for both Gaussian and Epanechnikov kernels are chosen using the rule of thumb suggested by Bowman and Azzalini[58], specifically

$$\mathbf{b} = \left(\frac{4}{N(d+2)}\right)^{1/(d+4)} \frac{\tilde{\boldsymbol{\sigma}}}{0.6745}. \qquad (20)$$

Here $N$ denotes the length of the data, and $\tilde{\boldsymbol{\sigma}}$ is the median absolute deviation estimator computed by $\tilde{\sigma} = \sqrt{\text{median}\{|v - \text{median}(v)|\}\text{median}\{|\mathbf{x} - \text{median}(\mathbf{x})|\}}$, where $v$ is the magnitude of the velocities. In general the bandwidth will go to zero with increasing data length, so the kernel estimator should be asymptotically unbiased. In that limit of infinite data, the differences between histogram and kernel density estimates are insignificant. When data is limited, we find the fastest convergence by using kernel density estimates with a multivariate Gaussian for $K$ and the Epanechnikov kernel for $L$.

To optimally handle limited data, the bandwidth is typically chosen to minimize the mean squared error (MSE) of the estimated function[59–61]:

$$\text{MSE}_{\hat{S}_{ss}} = \left\langle \left(\hat{S}_{ss} - \dot{S}_{ss}\right)^2 \right\rangle \text{ and } \text{MSE}_{TUR} = \left\langle \left(\hat{S}_{TUR} - \dot{S}_{TUR}\right)^2 \right\rangle, \qquad (21)$$

where the expectation value is taken over realizations of trajectories. The MSE is naturally a function of the bandwidth since the value of the estimator depends on **b**. Supplementary Figure 1 shows this bandwidth-dependence of the MSE estimated from the five-bead model temporal estimator and TUR lower bound with $\tau_{obs} = 1200$ and $T_c/T_h = 0.1$. Notice that the TUR lower bound tends to be less sensitive to the choice of bandwidth.

**Estimation of the TUR lower bound**. To get estimates for the current's mean and variance, $\widehat{\langle j_d \rangle}$ and $\widehat{\text{Var}(j_d)}$, from a single realization of length $\tau_{obs}$, we first divide the trajectory into $\tau_{obs}/\Delta\tau$ subtrajectories of length $\Delta\tau$. For the continuous-time

Markov jump process as shown in Fig. 3b, the vector field $\mathbf{d}(\mathbf{x})$ is discretized as a set of weights $d_{\mathbf{x}+\mathbf{h},\mathbf{x}}$ associated with the edges of the lattice and the trajectory is series of lattice sites occupied over time. The accumulated current, as illustrated in Fig. 3d, is computed as the sum of weights along the subtrajectory $k$: $J_{\mathbf{d}}^{(k)} = \sum_i \mathbf{d}_{\mathbf{x}_i,\mathbf{x}_{i+1}}$. For the continuous-space Langevin dynamics, the accumulated current for subtrajectory is given by $J_{\mathbf{d}}^{(k)} = \sum_i \mathbf{d}\left(\frac{\mathbf{x}_{i\Delta t}+\mathbf{x}_{(i-1)\Delta t}}{2}\right) \cdot \left(\mathbf{x}_{i\Delta t} - \mathbf{x}_{(i-1)\Delta t}\right)$. This accumulated current is scaled by the trajectory length to get the fluctuating macroscopic current for subtrajectory $k$: $j_{\mathbf{d}}^{(k)} = J_{\mathbf{d}}^{(k)}/\Delta\tau$. The sample mean and variance of $\left\{j_{\mathbf{d}}^{(1)}, j_{\mathbf{d}}^{(2)}, ...\right\}$ give $\widehat{\langle j_{\mathbf{d}}\rangle}$ and $\widehat{\mathrm{Var}(j_{\mathbf{d}})}$, respectively.

**Computing the mean and variance by tilting**. It is useful to conceptualize $\langle j_{\mathbf{d}}\rangle$ and $\mathrm{Var}(j_{\mathbf{d}})$ in terms of sampled trajectories, but finite trajectory sampling will result in statistical errors. We may alternatively compute the mean and variance as the first two derivatives of the scaled cumulant generating function $\phi(\lambda) = \lim_{\tau_{\mathrm{obs}}\to\infty} \frac{1}{\tau_{\mathrm{obs}}} \ln\langle e^{\lambda j_{\mathbf{d}}\tau_{\mathrm{obs}}}\rangle$, evaluated at $\lambda = 0$. The expectation value averages over all trajectories of length $\tau_{\mathrm{obs}}$, and in the long-time limit, $\phi(\lambda)$ coincides with the maximum eigenvalue of the tilted operator with matrix elements $\mathbb{W}(\lambda)_{\mathbf{x}+\mathbf{h},\mathbf{x}} = \mathbb{W}_{\mathbf{x}+\mathbf{h},\mathbf{x}} e^{\lambda d_{\mathbf{x}+\mathbf{h},\mathbf{x}}}$[54–56]. By discretizing space, we computed $\phi(\lambda)$ around $\lambda = 0$ as the maximal eigenvalue of the tilted operator. Using numerical derivatives, we estimate

$$\langle j_{\mathbf{d}}\rangle = \phi'(0) \approx \frac{\phi(\delta\lambda) - \phi(-\delta\lambda)}{2\delta\lambda} \qquad (22)$$

$$\mathrm{Var}(j_{\mathbf{d}}) = \phi''(0) \approx \frac{\phi(\delta\lambda) + \phi(-\delta\lambda)}{\delta\lambda^2} \qquad (23)$$

with $\delta\lambda = 0.00001$.

**MC optimization**. We seek a vector field $\mathbf{d}(\mathbf{x})$ such that the TUR bound is as large as possible. To identify such a choice of $\mathbf{d}$, we first decompose it into a basis of $M = 100$ Gaussians:

$$\mathbf{d}(\mathbf{x}) = \sum_{i=1}^{M} w^{(i)} \exp\left[(\mathbf{x} - \mathbf{x}^{(i)})B^{-1}(\mathbf{x} - \mathbf{x}^{(i)})\right]. \qquad (24)$$

The $i$th Gaussian, centered at position $\mathbf{x}^{(i)}$, carries a weight $w^{(i)}$. The centers for the first 50 Gaussians are uniformly sampled with $x_1$ ranging from $-50$ to 50 and $x_2$ from $-20$ to 20. The breadth of the Gaussians along the $i$ direction, $B_{ii}$, is set to 10% of the length of the interval from which uniform samples are drawn. Only the weights for these 50 Gaussians will be allowed to freely vary. The remaining 50 Gaussians are paired with the first 50 to impose the antisymmetry $\mathbf{d}(\mathbf{x}) = -\mathbf{d}(-\mathbf{x})$. Practically, this antisymmetry constraint is achieved by placing a second Gaussian at $-\mathbf{x}$ with the opposite weight as the Gaussian positioned at $\mathbf{x}$.

With this regularization, we replace the optimization of $\mathbf{d}$ with a sampling problem. We sample the first 50 weights $\mathbf{w}$ in proportion to $\exp(\beta \dot{S}_{\mathrm{TUR}}^{(\mathbf{d})})$, where $\beta$ is an effective inverse temperature and $\dot{S}_{\mathrm{TUR}}^{(\mathbf{d})}$ depends on the weights since $\mathbf{d}$ depends on $\mathbf{w}$. By choosing $\beta = 5000$, the sampling is strongly biased toward weights that give a near-optimal value of the TUR bound. After initializing the weights with uniform random numbers from $[-1, 1]$, Monte Carlo moves $\mathbf{w}\to\mathbf{w}'$ were proposed by perturbing the $w_i$'s by random uniform numbers drawn from $[-0.5, 0.5]$. The $\mathbf{d}'$ corresponding to these new weights was computed according to Eq. (24), and the TUR bound for that proposed macroscopic current was computed using numerical derivatives of the tilted operator $\mathbb{W}(\lambda)$ around $\lambda = 0$ as described above. The maximum eigenvalue calculations made use of Mathematica's implementation of the Arnoldi method, performed using sparse matrices. Each proposed move to $\mathbf{w}'$ was accepted with the Metropolis criterion $\min[1, \exp(-\beta(\dot{S}_{\mathrm{TUR}}^{(\mathbf{d})} - \dot{S}_{\mathrm{TUR}}^{(\mathbf{d}')}))]$.

In addition to starting from a random choice of $\mathbf{d}$, we performed MC sampling about the thermodynamic force by expressing $\mathbf{d}$ as

$$\mathbf{d}(\mathbf{x}) = \mathbf{F}(\mathbf{x}) + \sum_{i=1}^{M} w^{(i)} \exp\left[(\mathbf{x} - \mathbf{x}^{(i)})B^{-1}(\mathbf{x} - \mathbf{x}^{(i)})\right]. \qquad (25)$$

Again, we have 100 Gaussians, half of them uniformly placed throughout the space and the rest positioned to make the perturbation antisymmetric. We stochastically update the weights by adding a uniform random number drawn from $[-0.05, 0.05]$, and conditionally accept the update with the same Metropolis factor as before. The resulting TUR lower bound tends toward higher values until it hits a plateau (Fig. 5 blue line). For each temperature ratio in Fig. 4a, the MC sampling was run for 500 steps, after which the TUR bound achieved a plateau and further optimization is either impossible or at least significantly more challenging.

## Data availability
Representative data generated from sampling trajectories with the aforementioned codes can be accessed online at https://doi.org/10.5281/zenodo.2576526.

## Code availability
Computer codes implementing all simulations and analyses described in this manuscript are available for download at https://doi.org/10.5281/zenodo.2576526.

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

## Acknowledgements

We gratefully acknowledge the Gordon and Betty Moore Foundation for supporting T.R.G. and J.M.H. as Physics of Living Systems Fellows through Grant GBMF4513. This research was supported by a Sloan Research Fellowship (to N.F.), the J.H. and E.V. Wade Fund Award (to N.F.), and the Human Frontier Science Program Career Development Award (to N.F.).

## Author contributions

J.M.H, T.R.G. and N.F. designed research, J.L. and T.R.G. performed research and analyzed data, J.L., J.M.H., T.R.G. and N.F. wrote the paper.
