## [peer review file · Nature Communications]

Reviewers' comments:

Reviewer #1 (Remarks to the Author):

see attached

Reviewer #2 (Remarks to the Author):

The paper by Li discusses different ways of estimating entropy production rates applied to a beads-springs model in a temperature gradient. The paper explores three different ways of determining entropy production rates: by taking spatial averages, temporal averages and the lower bound provided by the thermodynamic uncertainty relation. The estimators are tested in numerical simulations of the aforementioned spring model. As such this paper is mostly methodological and potentially applicable to available experimental data. However no such tested is made on real experimental data so it is hard to evaluate its impact. I have reservations this paper will attract the interest of a broad readership. The paper is rather technical and the conclusions not so exciting. To my regret I cannot recommend it for publication.

Reviewer #3 (Remarks to the Author):

In this paper, the authors demonstrate a protocol for measuring dissipation rates in nonequilibrium systems by considering the exactly solvable bead-spring-coupled-to-hot/cold-reservoirs model. Apart from the obvious spatial and temporal averaged entropy production rate, they also consider the less obvious lower bound of the entropy production rate given by the thermodynamic uncertainty relations. Using numerical simulations of the model they measure the entropy production rates using the three possibilities in the case of two bead and five bead systems and establish convergence properties of these estimates as function of driving rate and dimensionality. Further, they consider optimization of the weight function that enters the bound given by the thermodynamic uncertainty relation.

Pros for the paper: The paper is a carefully considered work of potentially great relevance to various nonequilibrium systems that are presently the topic of extensive investigations. For an expert theorist, it is a very well written exposition of the key ideas. I especially liked learning about the weak driving limit and the fact that as dimensions go up I am much better off using the TUR bound.

Cons for the paper: I do not think the paper is accessible to the wide audience that Nat. Comm has and in particular to the many of us lay people who are not steeped in stochastic thermodynamics but need to understand this stuff in order we be able to apply this to our systems.

I realize the technical nature of the work limits what can be done for the accessibility of the paper. But I have a few suggestions:

a) I really liked the clear story laid out that leads the reader to Eq.7. I would suggest that a similar narrative should be put in just above and just below Eq. 14 so that the TUR does not just pop out of nowhere for the reader. I did not understand what $d(x)$ was till I went back and read some of the associated literature.

b) Expand the figure captions to hold the hand of the reader more. For example Fig 3 a. Adding one more sentence or a phrase to help the reader keep track of the fact that entropy production is a $j \cdot d$ thing and hence the statement of accumulation of current implies higher entropy production and therefore a “farther from equilibrium” conclusion. An inset in fig 3a that zooms in to show what $j \cdot d$ looks like. I am just giving some example here. The authors are best able to see what might help the reader follow along and realize the importance of what the authors are saying.

In conclusion, this referee really likes the paper and thinks it should be published following some revisions from the authors to enhance accessibility to non-experts.

Dear Dr. Dubrovina and Reviewers:

Here are point by point responses to the reviews.

Regards,

Junang Li, Jordan M. Horowitz, Nikta Fakhri, and Todd R. Gingrich

Review 1: Entropy production is a hallmark of non-equilibrium processes. How to determine it from experimental data is a major challenge especially when not all relevant degrees of freedom are accessible. Recently, a new theoretical tool, the thermodynamic uncertainty relation (TUR) has been found. It allows to infer a lower bound on entropy production from mean and variance of any (even a coarse-grained) current. The present authors discuss estimators based on different choices of the current in a case study for an analytically solvable model with two degrees of freedom. They also compare the estimate based on the TUR with a direct evaluation taking into account spatial and temporal coarse-graining. In general, I would not consider such a, in principle rather simple, case study as appropriate for a high profile journal. However, since the TUR has generated enormous interest in the community, a critical assessment on its practical implementation as done here may indeed find a broader audience and thus may have significant impact.

We agree that the model is simple and that the analysis rises to the level of this journal due to intense recent interest in making use of the TUR. We think a critical evaluation of simple models is particularly impactful when it can inform such experimental applications.

Before reaching a final recommendation, I would like to see a revised version addressing the following issues:

1. I did not understand the explanation given in the paragraph following eqs, 12, 13 (“Unsurprisingly...”). Phase space points that are visited “infrequently” will contribute only little to the estimate in eq. 12 since in these regions j will be small as well. So an error on F in those regions should not harm too much.

In our initial submission, we focused on the errors in F that are introduced by extrapolating with a small finite trajectory. As the reviewer points out, the spatial calculation uses estimates of F and of j , and estimates of j should be zero in the regions of phase space that have not been visited. Unfortunately, this is not quite correct, because the estimates of j are impacted (through a kernel function) by neighboring regions of phase space. Consequently, the kernel density estimation procedure can extrapolate noisy (and erroneous) estimates of both F and j in regions of phase space that have been inadequately sampled (or not yet sampled at all).

Because this point could be unnecessarily confusing, we revised the relevant section of the manuscript in a way we think is clearer:

“At first glance $\hat{S}_{ss}^{\text{spat}}$ and $\hat{S}_{ss}^{\text{temp}}$ may appear equivalent due to ergodicity. Indeed, with an infinite amount of sampling, both schemes must yield the same result, \dot{S}_{ss} , but the temporal estimator converges significantly faster with finite sampling. Plots of the estimated local dissipation rate (Fig. 2 inset) hint at the reason $\hat{S}_{ss}^{\text{spat}}$ converges more slowly: $\dot{\sigma}_{ss}(\mathbf{x})$ must be accurately estimated by $\hat{\sigma}_{ss}(\mathbf{x}) = \hat{F}(\mathbf{x}) \cdot \hat{j}(\mathbf{x})$ throughout the entire configuration space. The integral in Eq. (12) equally weights $\hat{\sigma}_{ss}(\mathbf{x})$ at all \mathbf{x} , even those points which have been infrequently (or never) visited by the stochastic trajectory. Our \mathbf{x} has dimension two, but we will also consider higher dimensional configuration spaces, for example by coupling more than two beads in a linear chain. If that configuration space has dimension greater than three or four, it becomes impractical to estimate $\dot{\sigma}$ across the entire space. Furthermore, estimating Eq. (12) for high-dimensional \mathbf{x} confronts the classic problem of performing numerical quadrature on a

high-dimensional grid, where it is well known that Monte Carlo integration becomes a superior method.

The temporal integral can be thought of as a convenient way to implement such a Monte Carlo integration, with sampled \mathbf{x} 's coming from the configurations of the stochastic trajectory. Notably, Eq. (13) is computed from estimates of the thermodynamic force near the sampled configurations $\mathbf{x}_{i\Delta t}$, precisely where the finite trajectory has most reliably sampled. In contrast, Eq. (12) requires spurious extrapolation of the kernel density estimates ($\hat{\rho}$ and $\hat{\mathbf{j}}$) to points which are far from the any sampled configurations. The advantage of the temporal estimator over the spatial one becomes even more pronounced as dimensionality increases. Nevertheless, even $\hat{S}_{ss}^{\text{temp}}$ becomes harder to estimate when \mathbf{x} grows in dimensionality. Getting accurate estimates of \mathbf{F} around the $\mathbf{x}_{i\Delta t}$ requires observing several trajectories which have cut through that part of configuration space while traveling in each direction. But when the dimensionality is large, recurrence to the same configuration-space neighborhood takes a long time. Consequently, we turn to a complementary method which can be informative even when \mathbf{x} is too high-dimensional to accurately estimate \mathbf{F} ."

2. The whole comparison between the direct estimates based on eq 12 or 13 and the one based on the TUR using $\mathbf{d}=\mathbf{F}$ is a little bit misguided. If one has access to the thermodynamic force \mathbf{F} , there is no need to use the TUR since then the integrated product of \mathbf{j} with \mathbf{F} gives the entropy production rate (up to discretization errors) whereas as the bound from the TUR needs not to be saturated.

We agree that if one has access to the thermodynamic force the thing to do is to estimate dissipation using the temporal estimator. Given the force, there is no reason to appeal to the TUR when the actual dissipation rate (as opposed to a bound) can be found directly. Our focus in this manuscript is the scenario in which one cannot get a clear picture of the thermodynamic force because the phase space is too vast. Then one could try to use the TUR to get a bound, making it important to understand how tight that bound will be. The tightness, however, is strongly influenced by which scalar macroscopic current is chosen (the choice of \mathbf{d}).

Our manuscript has considered two different ways to select \mathbf{d} : (1) as $\hat{\mathbf{F}}$, the potentially very noisy estimate for the thermodynamic force and (2) as a vector field which emerges from a Monte Carlo optimization procedure of the TUR ratio. Since (2) is generated so as to produce the largest value of the TUR ratio (the tightest bound), it may appear misguided to ever attempt (1). The problem is that (2) can suffer from over-fitting if one is not careful. By searching over the vast space of all possible choices for \mathbf{d} , one should expect (2) to return the best \mathbf{d} for the specific sampled data set, which could return erroneous over-fitted results. One way to handle over-fitting is to perform cross-validation, training the choice of \mathbf{d} based on one data set, but using it for a second data set. Selecting \mathbf{d} with (1) is an alternative to this cross validation that removes the risk of over-optimization. The reviewer's point is taken that if $\hat{\mathbf{F}}$ is estimated well it would be better to use the temporal estimator than the TUR strategy, but the bottom right inset to Fig. 4 show that there is a regime in which the TUR bound is tight and the TUR estimator convergence is noticeably faster than estimating with the temporal average. In this regime, $\hat{\mathbf{F}}$ is not a great estimate of \mathbf{F} , but it nevertheless offers a useful generalized current for TUR estimation while avoiding the over-optimization problem.

We have sought to address these issues in several different places. We have revamped the final two paragraphs of the section "Indirect Strategy for Entropy Production Inference," where the idea of choosing $\mathbf{d} = \mathbf{F}$ was first introduced. We have also edited the discussion section to better reflect the pros and cons of various choices of \mathbf{d} .

3. Overall, due credit is given to earlier work. Still, I noticed a few omissions that the authors may want to consider.

Thank you. We tried very hard to distribute credit where due, but of course there are always oversights. We very much appreciate these good suggestions.

3a. One striking example that shows the power of the TUR to bound the efficiency of molecular motors is given by Seifert in *Physica A*, 504, 176, 2018, where extant experimental data on kinesin are analyzed.

Yes, the kinesin analysis is an important application of the TUR to experimental data. We have made reference to the work shortly after Eq. (15), where we highlight how important it is that the current fluctuations are in a one dimensional space such that the mean and variance of the current can be well estimated by experiments.

3b. Entropy production in electric circuits that can be mapped on this bead-spring model has been investigated earlier theoretically and experimentally by Ciliberto's group, PRL 110, 180601, 2013.

Indeed, this is an important and very relevant reference. We have included in "Due to its analytic and experimental tractability, this bead-spring system and related variants have been extensively studied as models for nonequilibrium dynamics [34-39]."

3c. When introducing the expressions for thermodynamic force and entropy production for Langevin dynamics in the paragraph after eq. 6, they quote Ref. 39. In fact, these relations have been derived and discussed much earlier, e.g., by Qian in PRE 65, 016102, 2001.

Thank you. We have included the reference.

3d. An early estimate of entropy production based on experimentally measure currents has been performed in Ref. 27. While this paper is quoted once in a block of references in the introduction, it would seem fair to point out more prominently that the basic strategy discussed in the two sections in the main part on page 3 has been realized with experimental data there before. In fact, Ref. 27 preceded the nice, here prominently quoted, work from the collaboration around Schmidt, MacKintosh and Broedersz.

Thank you, this is a very good suggestion. That section of the introduction now reads:

"We set out to develop and explore strategies for inferring the dissipation rate from these experimentally-accessible nonequilibrium fluctuations. When all relevant degrees of freedom are tracked, Lander et al. have used interacting driven colloids to indirectly measure dissipation from fluctuations [27], but it should also be possible to bound dissipation on the basis of nonequilibrium fluctuations in a subset of the relevant degrees of freedom. As a tangible example of our motivation, consider the experiment of Battle et al., which tracked cilia shape fluctuations to determine that the cilia dynamics were driven out of equilibrium [9]."

Review 2: The paper by Li discusses different ways of estimating entropy production rates applied to a beads-springs model in a temperature gradient. The paper explores three different ways of determining entropy production rates: by taking spatial averages, temporal averages and the lower bound provided by the thermodynamic uncertainty relation. The estimators are tested in numerical simulations of the aforementioned spring model. As such this paper is mostly methodological and potentially applicable to available experimental data. However no such tested is made on real experimental data so it is hard to evaluate its impact. I have reservations this paper will attract the interest of a broad readership. The paper is rather technical and the conclusions not so exciting. To my regret I cannot recommend it for publication.

While we appreciate the feedback, we respectfully disagree with Reviewer 2. We believe that the topics are particularly timely (as evidenced by the excitement of Reviewers 1 and 3 and from our interactions with both theoretical and experimental colleagues.

Review 3: In this paper, the authors demonstrate a protocol for measuring dissipation rates in nonequilibrium systems by considering the exactly solvable bead-spring-coupled-to-hot/cold-reservoirs model. Apart from the obvious spatial and temporal averaged entropy production rate, they also consider the less obvious lower bound

of the entropy production rate given by the thermodynamic uncertainty relations. Using numerical simulations of the model they measure the entropy production rates using the three possibilities in the case of two bead and five bead systems and establish convergence properties of these estimates as function of driving rate and dimensionality. Further, they consider optimization of the weight function that enters the bound given by the thermodynamic uncertainty relation.

Pros for the paper: The paper is a carefully considered work of potentially great relevance to various nonequilibrium systems that are presently the topic of extensive investigations. For an expert theorist, it is a very well written exposition of the key ideas. I especially liked learning about the weak driving limit and the fact that as dimensions go up I am much better off using the TUR bound.

We appreciate this reviewer's positive feedback. We're glad that the work was generally received as well-written and we've sought to make the technical part of this work more accessible to an audience that does not entirely consist of expert theorists.

Cons for the paper: I do not think the paper is accessible to the wide audience that Nat. Comm has and in particular to the many of us lay people who are not steeped in stochastic thermodynamics but need to understand this stuff in order we be able to apply this to our systems. I realize the technical nature of the work limits what can be done for the accessibility of the paper. But I have a few suggestions: a) I really liked the clear story laid out that leads the reader to Eq. 7. I would suggest that a similar narrative should be put in just above and just below Eq. 14 so that the TUR does not just pop out of nowhere for the reader. I did not understand what $d(x)$ was till I went back and read some of the associated literature.

b) Expand the figure captions to hold the hand of the reader more. For example Fig 3 a. Adding one more sentence or a phrase to help the reader keep track of the fact that entropy production is a $j \cdot d$ thing and hence the statement of accumulation of current implies higher entropy production and therefore a farther from equilibrium conclusion. An inset in fig 3a that zooms in to show what $j \cdot d$ looks like. I am just giving some example here. The authors are best able to see what might help the reader follow along and realize the importance of what the authors are saying.

We're glad the narrative up through Eq. 7 was clear, and we strive to make the TUR section equally clear.

To this end, we have restructured the text between Eqs. (14) and (15). Even more importantly, we have incorporated a new Figure 3 (which was previously in the SI). This new figure zooms in on parts (a) and (b) of the old Figure 3 to illustrate how the empirical current j_d is constructed from microscopic transitions. We believe this new figure, with its caption, should help tremendously in pictorially showing how data from trajectories translates into the quantities $\langle j_d \rangle$ and $\text{Var}(j_d)$ that appear in Eq. (15).

Additionally, all of the captions have been expanded and improved, most notably the captions for Figure 3, the new Figure 4 (previously part of Fig. 3 and part of an SI figure), and the new Figure 5 both have improved captions.

Throughout the second half of the paper we have made several changes to cut, rephrase, or relegate to a footnote some of the more notationally dense and technical sections. As the reviewer points out, this is intrinsically a technical piece of work, so only so much can be done. We feel that real improvements have been made, however to the readability.

In conclusion, this referee really likes the paper and thinks it should be published following some revisions from the authors to enhance accessibility to non-experts.

Thank you! We'll be interested to learn if our rewritten sections have helped make the work more accessible.

Reviewer #1 (Remarks to the Author):

I have read the reply of the authors and the revised manuscript. I am fully satisfied with both

and do recommend publication as now is.

Reviewer #2 (Remarks to the Author):

I have seen the response by the authors to my concerns. However I cannot see in which direction the new submitted manuscript has improved. In their response to my criticism the authors respectfully disagree with my report. At least I was expecting to see an implementation of the method to experimental data or, in their absence, to numerical data obtained from a realistic model. The authors have not even made the least effort to show me that I am wrong. I do not see what my contribution as reviewer can be at this stage. I can only keep my recommendation of rejecting the manuscript.

Reviewer #3 (Remarks to the Author):

The revised manuscript indeed is more accessible to stochastic thermo novices. I am happy to recommend publication.

Dear Dr. Dubrovina and Reviewers:

Here are point by point responses to the reviews.

Regards,

Junang Li, Jordan M. Horowitz, Nikta Fakhri, and Todd R. Gingrich

Review 1: Entropy production is a hallmark of non-equilibrium processes. How to determine it from experimental data is a major challenge especially when not all relevant degrees of freedom are accessible. Recently, a new theoretical tool, the thermodynamic uncertainty relation (TUR) has been found. It allows to infer a lower bound on entropy production from mean and variance of any (even a coarse-grained) current. The present authors discuss estimators based on different choices of the current in a case study for an analytically solvable model with two degrees of freedom. They also compare the estimate based on the TUR with a direct evaluation taking into account spatial and temporal coarse-graining. In general, I would not consider such a, in principle rather simple, case study as appropriate for a high profile journal. However, since the TUR has generated enormous interest in the community, a critical assessment on its practical implementation as done here may indeed find a broader audience and thus may have significant impact.

We agree that the model is simple and that the analysis rises to the level of this journal due to intense recent interest in making use of the TUR. We think a critical evaluation of simple models is particularly impactful when it can inform such experimental applications.

Before reaching a final recommendation, I would like to see a revised version addressing the following issues:

1. I did not understand the explanation given in the paragraph following eqs, 12, 13 (“Unsurprisingly...”). Phase space points that are visited “infrequently” will contribute only little to the estimate in eq. 12 since in these regions j will be small as well. So an error on F in those regions should not harm too much.

In our initial submission, we focused on the errors in F that are introduced by extrapolating with a small finite trajectory. As the reviewer points out, the spatial calculation uses estimates of F and of j , and estimates of j should be zero in the regions of phase space that have not been visited. Unfortunately, this is not quite correct, because the estimates of j are impacted (through a kernel function) by neighboring regions of phase space. Consequently, the kernel density estimation procedure can extrapolate noisy (and erroneous) estimates of both F and j in regions of phase space that have been inadequately sampled (or not yet sampled at all).

Because this point could be unnecessarily confusing, we revised the relevant section of the manuscript in a way we think is clearer:

“At first glance $\hat{S}_{ss}^{\text{spat}}$ and $\hat{S}_{ss}^{\text{temp}}$ may appear equivalent due to ergodicity. Indeed, with an infinite amount of sampling, both schemes must yield the same result, \dot{S}_{ss} , but the temporal estimator converges significantly faster with finite sampling. Plots of the estimated local dissipation rate (Fig. 2 inset) hint at the reason $\hat{S}_{ss}^{\text{spat}}$ converges more slowly: $\dot{\sigma}_{ss}(\mathbf{x})$ must be accurately estimated by $\hat{\sigma}_{ss}(\mathbf{x}) = \hat{F}(\mathbf{x}) \cdot \hat{j}(\mathbf{x})$ throughout the entire configuration space. The integral in Eq. (12) equally weights $\hat{\sigma}_{ss}(\mathbf{x})$ at all \mathbf{x} , even those points which have been infrequently (or never) visited by the stochastic trajectory. Our \mathbf{x} has dimension two, but we will also consider higher dimensional configuration spaces, for example by coupling more than two beads in a linear chain. If that configuration space has dimension greater than three or four, it becomes impractical to estimate $\dot{\sigma}$ across the entire space. Furthermore, estimating Eq. (12) for high-dimensional \mathbf{x} confronts the classic problem of performing numerical quadrature on a

high-dimensional grid, where it is well known that Monte Carlo integration becomes a superior method.

The temporal integral can be thought of as a convenient way to implement such a Monte Carlo integration, with sampled \mathbf{x} 's coming from the configurations of the stochastic trajectory. Notably, Eq. (13) is computed from estimates of the thermodynamic force near the sampled configurations $\mathbf{x}_{i\Delta t}$, precisely where the finite trajectory has most reliably sampled. In contrast, Eq. (12) requires spurious extrapolation of the kernel density estimates ($\hat{\rho}$ and $\hat{\mathbf{j}}$) to points which are far from the any sampled configurations. The advantage of the temporal estimator over the spatial one becomes even more pronounced as dimensionality increases. Nevertheless, even $\hat{S}_{ss}^{\text{temp}}$ becomes harder to estimate when \mathbf{x} grows in dimensionality. Getting accurate estimates of \mathbf{F} around the $\mathbf{x}_{i\Delta t}$ requires observing several trajectories which have cut through that part of configuration space while traveling in each direction. But when the dimensionality is large, recurrence to the same configuration-space neighborhood takes a long time. Consequently, we turn to a complementary method which can be informative even when \mathbf{x} is too high-dimensional to accurately estimate \mathbf{F} ."

2. The whole comparison between the direct estimates based on eq 12 or 13 and the one based on the TUR using $\mathbf{d}=\mathbf{F}$ is a little bit misguided. If one has access to the thermodynamic force \mathbf{F} , there is no need to use the TUR since then the integrated product of \mathbf{j} with \mathbf{F} gives the entropy production rate (up to discretization errors) whereas as the bound from the TUR needs not to be saturated.

We agree that if one has access to the thermodynamic force the thing to do is to estimate dissipation using the temporal estimator. Given the force, there is no reason to appeal to the TUR when the actual dissipation rate (as opposed to a bound) can be found directly. Our focus in this manuscript is the scenario in which one cannot get a clear picture of the thermodynamic force because the phase space is too vast. Then one could try to use the TUR to get a bound, making it important to understand how tight that bound will be. The tightness, however, is strongly influenced by which scalar macroscopic current is chosen (the choice of \mathbf{d}).

Our manuscript has considered two different ways to select \mathbf{d} : (1) as $\hat{\mathbf{F}}$, the potentially very noisy estimate for the thermodynamic force and (2) as a vector field which emerges from a Monte Carlo optimization procedure of the TUR ratio. Since (2) is generated so as to produce the largest value of the TUR ratio (the tightest bound), it may appear misguided to ever attempt (1). The problem is that (2) can suffer from over-fitting if one is not careful. By searching over the vast space of all possible choices for \mathbf{d} , one should expect (2) to return the best \mathbf{d} for the specific sampled data set, which could return erroneous over-fitted results. One way to handle over-fitting is to perform cross-validation, training the choice of \mathbf{d} based on one data set, but using it for a second data set. Selecting \mathbf{d} with (1) is an alternative to this cross validation that removes the risk of over-optimization. The reviewer's point is taken that if $\hat{\mathbf{F}}$ is estimated well it would be better to use the temporal estimator than the TUR strategy, but the bottom right inset to Fig. 4 show that there is a regime in which the TUR bound is tight and the TUR estimator convergence is noticeably faster than estimating with the temporal average. In this regime, $\hat{\mathbf{F}}$ is not a great estimate of \mathbf{F} , but it nevertheless offers a useful generalized current for TUR estimation while avoiding the over-optimization problem.

We have sought to address these issues in several different places. We have revamped the final two paragraphs of the section "Indirect Strategy for Entropy Production Inference," where the idea of choosing $\mathbf{d} = \mathbf{F}$ was first introduced. We have also edited the discussion section to better reflect the pros and cons of various choices of \mathbf{d} .

3. Overall, due credit is given to earlier work. Still, I noticed a few omissions that the authors may want to consider.

Thank you. We tried very hard to distribute credit where due, but of course there are always oversights. We very much appreciate these good suggestions.

3a. One striking example that shows the power of the TUR to bound the efficiency of molecular motors is given by Seifert in *Physica A*, 504, 176, 2018, where extant experimental data on kinesin are analyzed.

Yes, the kinesin analysis is an important application of the TUR to experimental data. We have made reference to the work shortly after Eq. (15), where we highlight how important it is that the current fluctuations are in a one dimensional space such that the mean and variance of the current can be well estimated by experiments.

3b. Entropy production in electric circuits that can be mapped on this bead-spring model has been investigated earlier theoretically and experimentally by Ciliberto's group, PRL 110, 180601, 2013.

Indeed, this is an important and very relevant reference. We have included in "Due to its analytic and experimental tractability, this bead-spring system and related variants have been extensively studied as models for nonequilibrium dynamics [34-39]."

3c. When introducing the expressions for thermodynamic force and entropy production for Langevin dynamics in the paragraph after eq. 6, they quote Ref. 39. In fact, these relations have been derived and discussed much earlier, e.g., by Qian in PRE 65, 016102, 2001.

Thank you. We have included the reference.

3d. An early estimate of entropy production based on experimentally measure currents has been performed in Ref. 27. While this paper is quoted once in a block of references in the introduction, it would seem fair to point out more prominently that the basic strategy discussed in the two sections in the main part on page 3 has been realized with experimental data there before. In fact, Ref. 27 preceded the nice, here prominently quoted, work from the collaboration around Schmidt, MacKintosh and Broedersz.

Thank you, this is a very good suggestion. That section of the introduction now reads:

"We set out to develop and explore strategies for inferring the dissipation rate from these experimentally-accessible nonequilibrium fluctuations. When all relevant degrees of freedom are tracked, Lander et al. have used interacting driven colloids to indirectly measure dissipation from fluctuations [27], but it should also be possible to bound dissipation on the basis of nonequilibrium fluctuations in a subset of the relevant degrees of freedom. As a tangible example of our motivation, consider the experiment of Battle et al., which tracked cilia shape fluctuations to determine that the cilia dynamics were driven out of equilibrium [9]."

Review 2: The paper by Li discusses different ways of estimating entropy production rates applied to a beads-springs model in a temperature gradient. The paper explores three different ways of determining entropy production rates: by taking spatial averages, temporal averages and the lower bound provided by the thermodynamic uncertainty relation. The estimators are tested in numerical simulations of the aforementioned spring model. As such this paper is mostly methodological and potentially applicable to available experimental data. However no such tested is made on real experimental data so it is hard to evaluate its impact. I have reservations this paper will attract the interest of a broad readership. The paper is rather technical and the conclusions not so exciting. To my regret I cannot recommend it for publication.

While we appreciate the feedback, we respectfully disagree with Reviewer 2. We believe that the topics are particularly timely (as evidenced by the excitement of Reviewers 1 and 3 and from our interactions with both theoretical and experimental colleagues.

Review 3: In this paper, the authors demonstrate a protocol for measuring dissipation rates in nonequilibrium systems by considering the exactly solvable bead-spring-coupled-to-hot/cold-reservoirs model. Apart from the obvious spatial and temporal averaged entropy production rate, they also consider the less obvious lower bound

of the entropy production rate given by the thermodynamic uncertainty relations. Using numerical simulations of the model they measure the entropy production rates using the three possibilities in the case of two bead and five bead systems and establish convergence properties of these estimates as function of driving rate and dimensionality. Further, they consider optimization of the weight function that enters the bound given by the thermodynamic uncertainty relation.

Pros for the paper: The paper is a carefully considered work of potentially great relevance to various nonequilibrium systems that are presently the topic of extensive investigations. For an expert theorist, it is a very well written exposition of the key ideas. I especially liked learning about the weak driving limit and the fact that as dimensions go up I am much better off using the TUR bound.

We appreciate this reviewer's positive feedback. We're glad that the work was generally received as well-written and we've sought to make the technical part of this work more accessible to an audience that does not entirely consist of expert theorists.

Cons for the paper: I do not think the paper is accessible to the wide audience that Nat. Comm has and in particular to the many of us lay people who are not steeped in stochastic thermodynamics but need to understand this stuff in order we be able to apply this to our systems. I realize the technical nature of the work limits what can be done for the accessibility of the paper. But I have a few suggestions: a) I really liked the clear story laid out that leads the reader to Eq. 7. I would suggest that a similar narrative should be put in just above and just below Eq. 14 so that the TUR does not just pop out of nowhere for the reader. I did not understand what $d(x)$ was till I went back and read some of the associated literature.

b) Expand the figure captions to hold the hand of the reader more. For example Fig 3 a. Adding one more sentence or a phrase to help the reader keep track of the fact that entropy production is a $j \cdot d$ thing and hence the statement of accumulation of current implies higher entropy production and therefore a farther from equilibrium conclusion. An inset in fig 3a that zooms in to show what $j \cdot d$ looks like. I am just giving some example here. The authors are best able to see what might help the reader follow along and realize the importance of what the authors are saying.

We're glad the narrative up through Eq. 7 was clear, and we strive to make the TUR section equally clear.

To this end, we have restructured the text between Eqs. (14) and (15). Even more importantly, we have incorporated a new Figure 3 (which was previously in the SI). This new figure zooms in on parts (a) and (b) of the old Figure 3 to illustrate how the empirical current j_d is constructed from microscopic transitions. We believe this new figure, with its caption, should help tremendously in pictorially showing how data from trajectories translates into the quantities $\langle j_d \rangle$ and $\text{Var}(j_d)$ that appear in Eq. (15).

Additionally, all of the captions have been expanded and improved, most notably the captions for Figure 3, the new Figure 4 (previously part of Fig. 3 and part of an SI figure), and the new Figure 5 both have improved captions.

Throughout the second half of the paper we have made several changes to cut, rephrase, or relegate to a footnote some of the more notationally dense and technical sections. As the reviewer points out, this is intrinsically a technical piece of work, so only so much can be done. We feel that real improvements have been made, however to the readability.

In conclusion, this referee really likes the paper and thinks it should be published following some revisions from the authors to enhance accessibility to non-experts.

Thank you! We'll be interested to learn if our rewritten sections have helped make the work more accessible.